# FLNC Expression Level Influences the Activity of TEAD-YAP/TAZ Signaling

**DOI:** 10.3390/genes11111343

**Published:** 2020-11-13

**Authors:** Anastasia Knyazeva, Aleksandr Khudiakov, Raquel Vaz, Aleksey Muravyev, Ksenia Sukhareva, Thomas Sejersen, Anna Kostareva

**Affiliations:** 1Almazov National Medical Research Centre, 197341 Saint-Petersburg, Russia; hudiakov.aa@gmail.com (A.K.); myravyoval@mail.ru (A.M.); k.sukhareva@gmail.com (K.S.); anna.kostareva@ki.se (A.K.); 2Department of Molecular Medicine and Surgery and Center for Molecular Medicine, Karolinska Institute, 171 76 Stockholm, Sweden; raquel.vaz@ki.se; 3Graduate School of Life and Health Science, University of Verona, 10 37134 Verona, Italy; 4Department of Women’s and Children’s Health, Karolinska Institute, 171 77 Stockholm, Sweden; thomas.sejersen@ki.se

**Keywords:** FLNC, genome editing, myogenic differentiation, Hippo signaling

## Abstract

Filamin C (FLNC), being one of the major actin-binding proteins, is involved in the maintenance of key muscle cell functions. Inherited skeletal muscle and cardiac disorders linked to genetic variants in *FLNC* have attracted attention because of their high clinical importance and possibility of genotype-phenotype correlations. To further expand on the role of FLNC in muscle cells, we focused on detailed alterations of muscle cell properties developed after the loss of FLNC. Using the CRISPR/Cas9 method we generated a C2C12 murine myoblast cell line with stably suppressed *Flnc* expression. FLNC-deficient myoblasts have a significantly higher proliferation rate combined with an impaired cell migration capacity. The suppression of *Flnc* expression leads to inability to complete myogenic differentiation, diminished expression of *Myh1* and *Myh4*, alteration of transcriptional dynamics of myogenic factors, such as *Mymk* and *Myog*, and deregulation of Hippo signaling pathway. Specifically, we identified elevated basal levels of Hippo activity in myoblasts with loss of FLNC, and ineffective reduction of Hippo signaling activity during myogenic differentiation. The latter was restored by *Flnc* overexpression. In summary, we confirmed the role of FLNC in muscle cell proliferation, migration and differentiation, and demonstrated for the first time the direct link between *Flnc* expression and activity of TEAD-YAP\TAZ signaling. These findings support a role of FLNC in regulation of essential muscle processes relying on mechanical as well as signaling mechanisms.

## 1. Introduction

Filamin C (FLNC) is a member of the actin-binding filamin protein family expressed predominantly in skeletal and cardiac muscle tissue [1,2]. Pathogenic variants in this gene have long been associated with neuromuscular disorders such as distal myopathies and myofibrillar myopathies [3,4,5]. Recently, the *FLNC* gene was also implicated in a variety of inherited cardiac disorders, such as restrictive, hypertrophic, dilated and arrhythmogenic cardiomyopathies, often in association with a neuromuscular phenotype [6]. The phenotypic variability of filaminopathies can be attributed to several distinct molecular mechanisms triggered by *Flnc* mutations [7]. Missense mutations are predominantly associated with misfolded protein aggregate formation and are often linked to myofibrillar myopathies and restrictive cardiomyopathies, while truncating mutations leading to protein haploinsufficiency and loss of dimerization are mainly associated with dilated and arrhythmogenic cardiomyopathies [8,9,10,11,12].

The precise role of FLNC in muscle cell function and regulation of molecular pathways currently remains a subject for intensive research. It was reported that *Flnc* expression gradually increases during C2C12 myogenic differentiation, and during regeneration after skeletal muscle injury [13]. Knock-down of *Flnc* in mouse myoblast C2C12 cells leads to impaired formation of elongated myotubes under differentiation conditions (data obtained on a FLNC-deficient mouse model) and FLNC is considered to be critical for mouse viability and muscle development [14]. However, the detailed impact of FLNC on muscle cell functional and differentiation properties is still under discussion and the link between FLNC levels, actin cytoskeleton reorganization and signaling cascades activity in muscle cells have not yet been studied in detail.

Muscle cell properties and function are tightly linked to actin cytoskeleton remodeling and reorganization. Since all filamin family members (FLNA, FLNB and FLNC) are able to bind actin filaments through an N-terminal actin-binding domain and form orthogonal actin network [15,16], these proteins could regulate actin reorganization dependent processes such as cell migration, proliferation and differentiation in different types of cells [17]. Recent studies have identified Yes-associated protein 1 (YAP) and Transcriptional coactivator with PDZ-binding motif (TAZ), members of the Hippo signaling pathway, as regulators of myocyte proliferation and mechanotransduction, at least in part, through interaction with actin cytoskeleton [18,19,20]. We hypothesized that loss of FLNC in myoblast cells can lead to changes in cell migration, proliferation and muscle differentiation, as well as dysregulation of signaling cascades, such as Hippo pathway. Using C2C12 myoblast *Flnc* knock-out (*FlncKO*) cells we demonstrated an increased proliferative dynamics of *FlncKO* myoblasts, altered differentiation capacity and overactivation of Hippo signaling in dividing cells and under myogenic differentiation after the loss of FLNC. We confirmed the essential role of FLNC in myogenic differentiation and highlighted a tight link between FLNC levels and activity of the Hippo signaling pathway.

## 2. Material and Methods

### 2.1. Generation of Flnc Knock-Out Cell Line Using CRISPR/Cas9

Generation of the *Flnc* knock-out C2C12 murine myoblast cell line (*FlncKO*) was performed by targeting the first exon of the mouse *Flnc* gene (NC_000072.6 (2933153.29461889)) using a previously described protocol [21]. Design of CRISPR/Cas9 guide RNA sequences was performed using the mit.broadinsitute/crispr online tool [21]. GuideRNA sequence was chosen among others by highest specifity score and minimal predicted off-target effect. A scramble guide RNA which does not target any mouse genome fragments was chosen as a scramble-control (Appendix A) [22]. Guide RNA sequences were cloned into pSpCas9(BB)-2A-GFP (PX458) (#48138, Addgene, Watertown, MA, USA) plasmid using *BbsI* restriction sites. Generated plasmids were verified by Sanger sequencing with U6 primer (Appendix A). C2C12 cells were transfected with CRISPR plasmid using Lipofectamine 3000 (Invitrogen, Carlsbad, CA, USA) according to the manufacturer’s instructions. Twenty-four hours after transfection GFP-positive cells were sorted using a FACSAria flow cytometer (BD Biosciences, San Jose, CA, USA). After sorting, cells were seeded in 96-wells plates at a density of 33 cells/plate. Plates were screened under a light microscope and the individual clones were grown up and re-plated for subsequent freezing, genomic DNA analysis and protein quantification.

TA-cloning of derived *FlncKO* clone alleles was performed using a Quick-TA kit (Evrogen, Moscow, Russia). Primers for amplification of desired DNA fragments are described in Appendix A.

### 2.2. Cell Culture

C2C12 murine mouse myoblasts (CRL-1772, ATCC, Manassas, VA, USA) were cultured in Dulbecco’s Modified Eagle Medium (Gibco, Waltham, MA USA) supplemented with 10% fetal bovine serum (FBS) (Gibco, USA), 2 mM L-glutamine, 50 IU/mL penicillin and 50 μg/mL streptomycin, hereafter mentioned as growth media. Cells were passaged every two days. For muscle differentiation, cells were seeded at 70% confluency. The following day, media was replaced to differentiation media (DM), consisting of DMEM supplemented with 2% horse serum, 2 mM L-glutamine, 50 IU/mL penicillin and 50 μg/mL streptomycin. Differentiation media was replaced every day for the duration of the experiment.

### 2.3. Western Blotting

Cells were trypsinized, pelleted by centrifugation at 300 g for 5 min and lysed in RIPA buffer, containing 50 mM NaCl, 25mM Tris-HCl pH8.0, 0.1% SDS. Lysates were centrifuged at 16,000 g and Laemmli buffer was added to supernatant and further incubated for 5 min at 100 °C. Protein lysates were run in 6% polyacrylamide gel and transferred to 0.45 µm nitrocellulose membrane. Membranes were blocked with 5% milk in PBS with 0.05% Tween-20 (PBS-T) (Sigma Aldrich, St. Louis, MO, USA) for 30 min and incubated with an anti-FLNC primary antibody (NBP1-89300, Novus Biologicals, Littleton, CO, USA) or anti-β-actin primary antibodies (AC-15, ab6276, Abcam, Cambridge, UK) for 16 h. Membranes were washed 3 times in PBS-T and incubated with secondary antibodies (Immun-Star Goat Anti-Rabbit/Mouse (GAR/GAM) HRP Conjugate, Bio-Rad Hercules, CA, USA), 1:10000 dilution in 5% milk for 1 h. Chemiluminescence was detected after application of SuperSignal West Femto substrate (Thermo Fisher Scientific, Waltham, MA, USA). Images were analyzed using ImageJ software (NIH, Bethesda, MD, USA).

### 2.4. Immunofluorescence

Prior to staining, cells were fixed in 4% paraformaldehyde (Sigma Aldrich, USA) for 10 min. Cell membrane was permeabilized with 0.5% Triton X-100 for 5 min and cells were blocked with 15% FBS for 1 h. Cells were incubated with MyHC primary antibodies (MAB4470, R&D Systems, Minneapolis, MN, USA) diluted 1:200 in 15% FBS for 1 h at RT and washed three times with PBS for 5 min. Incubation with Goat anti-Mouse Secondary Antibodies, Alexa Fluor 488 (Cat # A-11001, Thermo Fisher Scientific, USA) diluted 1:1000 in PBS was performed for 45 min at RT. For staining nuclei 4′,6-diamidino-2-phenylindole (DAPI, Thermo Fisher Scientific, USA) was used at a concentration of 0.1 μg/mL. Fusion index was determined as the ratio of nuclei within each MyHC-positive myotubes to the total number of nuclei. A minimum of 2 nuclei per myotube was required for inclusion in the quantification. Analysis was performed for 10 separate fields. F-actin staining was performed using Rhodamine Phalloidin probe (Thermo Fisher Scientific, USA), according to manufacturer’s instructions.

### 2.5. Gene Expression Analysis

Cells were lysed in ExtractRNA reagent (Evrogen, Russia). Total RNA was extracted according to manufacturer’s instructions. After extraction, RNA was quantified using NanoDrop 3300 (ThermoFisher, USA). Complementary DNA (cDNA) was generated using random primers and MMLV reverse transcription kit (Evrogen, Russia). Quantitative measurement of gene expression was performed using qPCR with qPCR mix-HS SYBR+ROX (Evrogen, Russia). Sequences for qPCR primers are listed in Appendix A. Gene expression was normalized to *Gapdh* mRNA level. In differentiation experiments data for each cell line was normalized to undifferentiated condition (day 0).

### 2.6. Plasmid Construction and Flnc Overexpression

The pCS2*^FLNC^* plasmid was generated as described previously [6]. For FLNC overexpression in C2C12 myoblasts, cells were transfected with Lipofectamine 3000 (Invitrogen, Carlsbad, CA, USA) according to manufacturer’s instructions. GFP-positive signal and, therefore, FLNC were detected 24 h after transfection.

### 2.7. Scratch Assay

Cells were seeded at a high density in 24-well plates. The following day a vertical scratch was applied using a yellow 200-μL pipette tip and growth media was replaced to serum free DMEM, supplemented with 2 mM L-glutamine, 50 IU/mL penicillin and 50 μg/mL streptomycin, this time point was established as “0 h”. Pictures were taken every two hours until the 12-h time point. Percentage of scratch healing was quantified as (scratch area_0 h_—scratch area_X hours_)/scratch area_0 h_·100.

### 2.8. Proliferation and Migration Analyses

Proliferation and migration analyses were performed using a xCELLigence Real-Time Cell Analyzer (RTCA) (ACEA Biosciences, San Diego, CA, USA). For measurement of proliferation dynamics, cells were seeded in 200 µL of growth media at a density of 5000 cells/well into an E-plate 16 (ACEA Biosciences, USA). Proliferation curves were evaluated over the course of 72 h with measurements taken every 15 min. Doubling time was analyzed from 8 to 43 h using RTCA software 2.1.0 (ACEA Biosciences, USA). For measurement of migration dynamics, 15,000 cells/mL density were seeded in 100 µL of serum-free media in upper chambers of CIM-plates 16 (ACEA Biosciences). The lower chamber was filled with 160 µL of growth media as a chemoattractant. Cell migration was evaluated over the course of 24 h with measurements taken every of 15 min. Slopes of migration curves were analyzed from 15 to 23 h using RTCA software 2.1.0 (ACEA Biosciences, USA).

### 2.9. Luciferase Assay

For evaluation of Hippo-pathway activity, cells were seeded to 96-well white microplates and co-transfected with HOP-flash plasmid (#83467, Addgene) and pGL4.73 [hRluc/SV40] (Promega, Madison, WI, USA) plasmids using Lipofectamine 3000. For undifferentiated cells, assay was conducted 24 h after transfection. For differentiation experiments, myogenic differentiation was induced 24 h after transfection. Cell lysates preparation and substrate application was performed using the Dual-Glo Luciferase Assay System (Promega, USA). Luminescence was measured using Synergy2 spectrophotometer (BioTek, Winooski, VT, USA). Firefly luciferase activity was normalized to Renilla luciferase activity. As a negative control, cells were transfected with HIP-flash (#83466, Addgene) and hRLuc/SV40 plasmids.

### 2.10. Statistical Analysis

Statistically significant differences between groups were determined by parametric Student’s *t*-tests (two-tailed, unpaired) using GraphPad Prism software. *p*-value < 0.05 was considered significant.

## 3. Results

### 3.1. Generation and Characterization of FlncKO Cell Line

A *Flnc* knock-out C2C12 murine myoblast cell line (*FlncKO*) was generated using CRISPR/Cas9 technique by targeting the first exon of the mouse *Flnc* gene (Appendix A). The control cell line was generated by scramble CRISPR-editing of C2C12 cells. *Flnc* first exon sequencing and subsequent sequencing of separate alleles cloned into TA-vectors confirmed the presence of two deletions in both alleles (Appendix A). These deletions led to the stable loss of protein expression in undifferentiated C2C12 cells. Notably, upon differentiation, a weak expression of *Flnc* was induced in *FlncKO* cells, potentially due to use of an alternative start codon (Appendix A). Nevertheless, the FLNC levels in *FlncKO* undifferentiated myoblasts was more than 1000-fold lower compared to scramble control myoblasts (Appendix A). Thereafter, we characterized mRNA expression level of other filamin isoforms—*Flna* and *Flnb*. Levels of *Flna* and *Flnb* expression decrease during myogenic differentiation in control cell line as well as in *FlncKO* cell line, notably, with the similar fold changes (Appendix A, upper graphs). Compared to control scramble cell line, expression of *Flna* in *FlncKO* cells is elevated in basal conditions (not differentiated) as well as at the late point of differentiation (Appendix A, lower graphs). At the same time, *Flnb* expression remains unaltered when compared with the scramble control. For cytoskeleton characterization we evaluated actin filaments distribution in *Flnc* knock-out cells using phalloidin staining. When compared with the scramble control, *FlncKO* cells do not reveal drastic changes of actin filaments scaffold (Appendix A); however, numbers of filopodia are slightly increased (Appendix A). Therefore generated *FlncKO* C2C12 cell line could be applied as a model of stable loss of FLNC with moderate cytoskeleton rearrangements.

### 3.2. FlncKO Myoblasts Exhibit Increased Proliferation Dynamic and Reduced Migration Ability

To assess the proliferation dynamics of *FlncKO* C2C12 cells, the real-time migration analysis (RTCA) system was used. Proliferation curves indicated that control cells achieved the maximal density and switched to the plateau phase after 43 h after start of assay (Figure 1A). In contrast, *FlncKO* cells did not reach a plateau phase and continued to proliferate. Analysis of proliferation curves revealed a significantly shorter doubling time of *FlncKO* cells compared to scramble control, indicating an increase in proliferation rate (Figure 1B).

Cellular migration was investigated using scratch assay in serum-free conditions to avoid the impact of proliferation. Already 6 h after the initial scratch, *FlncKO* cells started to demonstrate decreased migration dynamic compared to control cells (Figure 1C,D). These experiments were further validated using an RTCA migration assay. For both cell lines, the slopes of migration curves were analyzed within 8 h (starting at 15 h after plating)—the time points considered to be optimal in between positive migration signals and achieving the plateau stage (Figure 1E). As shown in Figure 1F, the rate of migration was significantly lower in *FlncKO* cells, as compared to scramble control cells, based on calculated slopes reflecting the cell index curves. Overall, *FlncKO* murine myoblasts display higher proliferation dynamics and impaired migration, as compared to scramble control myoblasts.

### 3.3. FlncKO Cells are Characterized by Reduced Ability of Myogenic Differentiation

Skeletal muscle differentiation was induced in scramble control and FLNC-deficient C2C12 myoblasts by changing the growth medium to differentiation medium. Five days after induction of myogenic differentiation, cells were investigated by immunofluorescence analysis using an anti-Myosin Heavy Chain (MyHC) antibody. While control myoblasts extensively formed multinucleated myotubes, FLNC-deficient cells displayed poor fusion capacity, a significantly lower fusion index and only rarely demonstrated positivity for MyHC staining (Figure 2A,B).

We further confirmed reduced differentiation capacity by revealing reduced expression of the late stage myogenic markers such as *Myh1* and *Myh4* and muscle-specific fusion factor *Myomaker* (*Mymk*) (Figure 2C). Expression of *Mymk* and other genes involved in muscle development is maintained under control of myogenic regulatory factors (MRFs), such as *Myog* [23]. While control cells demonstrated a decline of *Myog* mRNA level upon differentiation completion, *Myog* remained elevated in *FlncKO* during the late stages of differentiation (Figure 2C). In summary, *FlncKO* cells display a deficient differentiation capacity and altered expression of analyzed myogenic differentiation markers.

### 3.4. Downregulation of FLNC in Myoblasts Promotes TEAD-YAP/TAZ Activation

In various cell types, including myoblasts, proliferation is maintained under control of the Hippo-signaling pathway and is associated with pro-proliferative gene expression activation. We monitored the Hippo pathway activity in *FlncKO* myoblasts, performing a luciferase TEAD-YAP/TAZ activity reporter assay. At low cell density, Hippo pathway activity was maintained at a low level both in control and *FlncKO* myoblasts, while at high cell density, Hippo became activated. However, *FlncKO* myoblasts demonstrated significantly stronger activation compared to the scramble control cell line (Figure 3A). Myogenic differentiation leads to a decrease in activity of Hippo-signaling 2 days after induction both in control and *FlncKO* myoblasts. Notably, this decrease was significantly less in *FlncKO* myoblasts than in control cells (2-fold compared to 5-fold) reflecting the inability of FLNC-deficient myoblasts to effectively downregulate Hippo signaling activity upon differentiation. Thus, in *FlncKO* myoblasts Hippo signaling remains activated on day 2 of myogenic differentiation compared to control the scramble control cell line. This observation was further supported by expression level of Hippo-dependent genes *Ctgf* (*Ccn1*) and *CyR61* (*Ccn2*). The decline of *Ctgf* mRNA level upon differentiation was 10-fold in control and only 2-fold in *FlncKO* cells. Similarly, we observed inefficient downregulation of *CyR61* (*Ccn2*) mRNA in *FlncKO* myoblasts compared to scramble control cells (Figure 3B). Thus, *FlncKO* myoblasts demonstrate an increased Hippo activity in proliferating cells and an inability to downregulate Hippo activity upon differentiation.

To further confirm the interconnection between *Flnc* expression and Hippo activity we performed rescue experiments using plasmid DNA (pCS2*^FLNC^*) encoding for human FLNC transcript. TEAD-YAP/TAZ activity reporter assay was assessed in scramble, *FlncKO* and restored *FlncKO* + pCS2*^FLNC^* proliferating and differentiating myoblasts. After FLNC rescue, TEAD-YAP/TAZ activity was significantly reduced compared to untransfected *FlncKO* myoblasts both in proliferating and in differentiated cells, thus confirming the dependence of Hippo-pathway activity on FLNC levels (Figure 3C).

## 4. Discussion

FLNC is an actin-binding protein which belongs to the filamin family and plays various structural and signaling roles in myocytes [1,2]. Being implicated in a broad range of neuromuscular and cardiac disorders, FLNC came into the scope of detailed cellular and molecular research. In the present report, we focused on FLNC functions in basic myocytes properties using a *Flnc* knock-out model generated by CRISPR/Cas9. Surprisingly, we did not observe critical changes of F-actin network in myoblast cells. Our data are in line with previously published data that loss of another filamin family protein, FLNA, in variety of cell lines is also not associated with F-actin defects [24]. We demonstrated that loss of FLNC leads to enhanced myoblast proliferation dynamics, and impairs key processes involved in myogenesis and functional muscle maturation. In contrast to the data obtained, *Flnc* knock down in cancer cells has been reported to lead to proliferation impairment, and high expression of FLNC is considered as predictive for tumorigenesis outcome [17,25]. This discrepancy could be explained by different roles of FLNC in contractile muscle cells and non-contractile cancer cells. The filamin proteins are also known for their ability to regulate cell motility and cell migration processes. Particularly, FLNC is considered as a sensor of cell-cell contact-dependent signaling due to its interaction with integrins and other proteins of the mechanosensory systems [26]. Here, we identified a reduced migration capacity in *FlncKO* myoblasts. Interestingly, the opposite phenomenon has been reported in cancer research with *Flnc* knock-down linked to increased migration and invasion in cancer cells [25,27]. The mechanism of FLNC effect on cell adhesion and migration could be more complex in muscle cells due to interaction of FLNC not only with sarcolemma integrin systems, but also with sarcomeric Z-disk structures, and thereby providing participation in mechanotransduction processes [28]. Loss of FLNC in mouse myocardium lead to upregulation of several adhesion proteins including b1D-integrin, Talin1, Kindlin2, Focal Adhesion Kinase, Vinculin and Integrin-Linked Kinase (ILK) [29]. Similarly, RNA-sequencing of cardiac samples of ARVC patients with FLNC loss of function variants revealed differential expression of genes, involved in cell adhesion, especially highlighting the importance of ILK signaling [30]. This link between structural sarcomeric gene mutations and cell adhesion proteins was also suggested in our previous study using a bioinformatic approach [31]. Together, these reports underscore the putative role of FLNC in migration and proliferation processes, possibly, through modulating the function of adhesion complexes in muscle cells.

The transition from a proliferative state to activation of differentiation pathways in muscle cells is under the control of several signaling cascades with the Hippo pathway being one of the major regulators of proliferation-differentiation switch [32,33]. In the present report, we demonstrated increased TEAD/YAP-TAZ activity in proliferating conditions of FLNC-deficient cells and the inability to downregulate its activity upon differentiation to the level similar to that of control cells. Rescue experiments support the fact, that Flnc expression level is tightly associated with Hippo signaling activity in myoblasts. In skeletal muscle, the Hippo pathway is a critical signaling pathway regulating cell proliferation and growth programs and its transcriptional co-factor Yes-associated protein (YAP) was shown to inhibit myogenic differentiation at a high dose [34]. In our established FLNC-deficient model myogenesis was completely interrupted, which is consistent with previous data acquired on *Flnc* knocked-down myoblasts derived using siRNA technique [14]. Both in wild-type and *FlncKO* myoblasts we observed substantial reduction of expression of two other filamin genes (*Flna* and *Flnb*) upon myogenic differentiation. Notably, we observed the increased expression of *Flna* in *Flnc* knocked-down myoblasts after differentiation, possibly, aimed on compensation of the lack of Flnc expression. The differentiation impairment was confirmed not only by morphological analysis but also by evaluation of transcriptional level of myogenic transcriptional factors Notably, besides the impaired upregulation of myotubes maturation markers *Myh1* and *Myh4* and fusion marker *Mymk*, we determined an abnormal expression dynamics of the transcriptional factor *Myog*, highlighting a dysregulation of the muscle differentiation program. This inability of FLNC-deficient cells to differentiate was associated with weak inactivation of Hippo during the myogenesis induction. Together, overall elevated Hippo activity in *FlncKO* cells and not completely suppressed expression of Hippo-targeted pro-proliferative genes give the reason to suggest sustained proliferative state even after differentiation induction. This supports earlier reports that in C2C12 cells Yap inhibition is essential for promoting the terminal differentiation from myoblasts to myotubes [35]. Thus, the role of FLNC in muscle maturation can be mediated through Hippo pathway activity. It is well known that in cardiomyocytes, disturbance of actin cross-linking and polymerization can be regulated by TEAD-YAP/TAZ activity and vice versa [20,36]. Whether or not downregulation of *Flnc* influences the Hippo pathway activity through modulation of actin cytoskeleton needs to be further elucidated.

FLNC still attracts high research interest due to new loss-of-function pathogenic isoforms identified in patients with severe form of myopathies and cardiomyopathies. Our data expand on the functions of FLNC in muscle cells towards its effect on migration, proliferation and differentiation. It uncovers the direct impact of FLNC absence on the Hippo pathway activity in proliferating and differentiating muscle cells. We confirmed the essential role of FLNC in the myogenic program implementation and highlighted a tight link between FLNC levels and activity of Hippo signaling.

## Figures and Tables

**Figure 1 genes-11-01343-f001:**
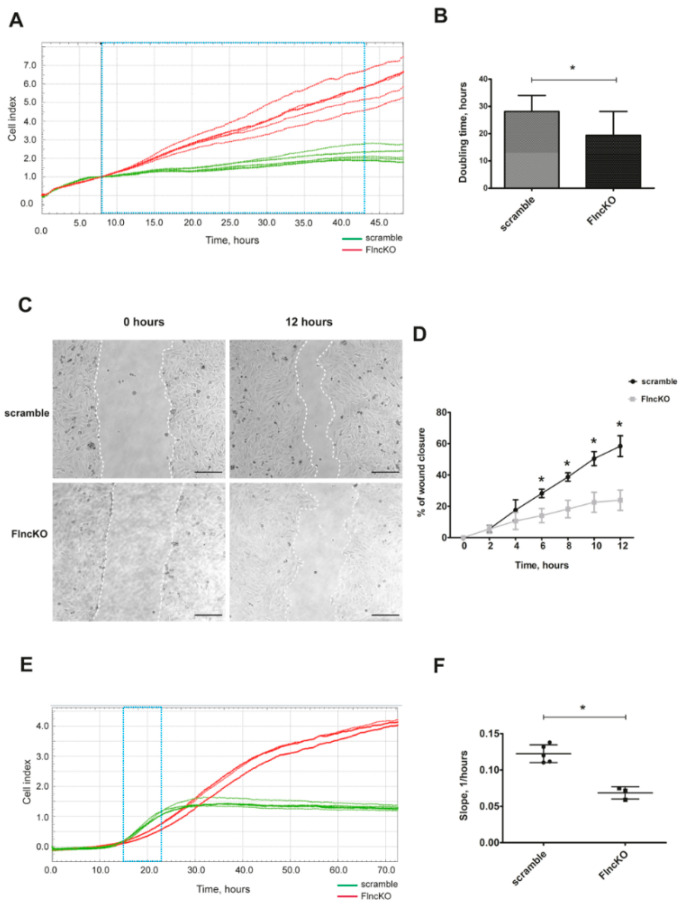
Reinforced proliferation and impairment of migration ability of *FlncKO* line. (**A**) Proliferation dynamics was obtained by xCELLigence Real-Time Cell Analyzer (RTCA), normalized cell index (CI) curves are presented. (**B**) Quantification of doubling time in 15 to 43 hours’ time range revealed an increase of proliferation rate in *FlncKO* cell line (*n* = 15 for each cell line). (**C**) Scratch assay shows a decrease of migration capacity from *FlncKO* cell line compared to scramble control; cell migration was recorded from 2 to 12 h. Representative images are shown for 12 hours’ time point. Dashed lines indicate area for quantification. (**D**) Reduction in cell migration was significant from 6 hours’ time point until 12 hours’ time point (*n* = 4 for each cell line and time points). (**E**) Validation of migration results performed by RTCA analysis. Migration curves represent cell index in each time point. (**F**). Quantification analysis of migration curves slopes was performed between 15 to 23 h after cell seeding. Scale bar 200 μM. For all * *p* < 0.05.

**Figure 2 genes-11-01343-f002:**
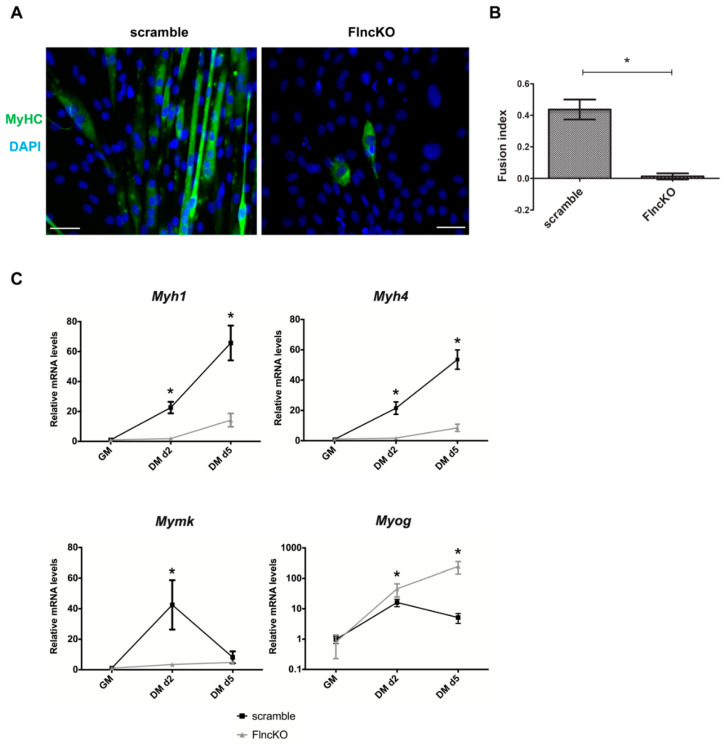
Diminished myogenic differentiation in FLNC-deficient C2C12 myoblasts. (**A**) FLNC deficiency dramatically reduces myoblast differentiation with complete inability to develop mature multinucleated myotubes 5 days after induction of myogenic differentiation in serum-depleted conditions. Only a small fraction *FlncKO* cells express MyHC, detected by immunofluorescence. (**B**) *FlncKO* cells shows a reduced fusion index, calculated after immunofluorescent staining of MyHC 5 days after induction of myogenic differentiation; *n* = 10 individual fields, in which fusion index analysis was performed. (**C**) Expression dynamics of muscle-specific markers are altered in *FlncKO* cells. Relative mRNA level of *Myh1*, *Myh4, Mymk* and *Myog* in *FlncKO* cells compared to scramble control both two and five days after muscle differentiation induction. Scale bar 50 μM. For all * *p* < 0.05.

**Figure 3 genes-11-01343-f003:**
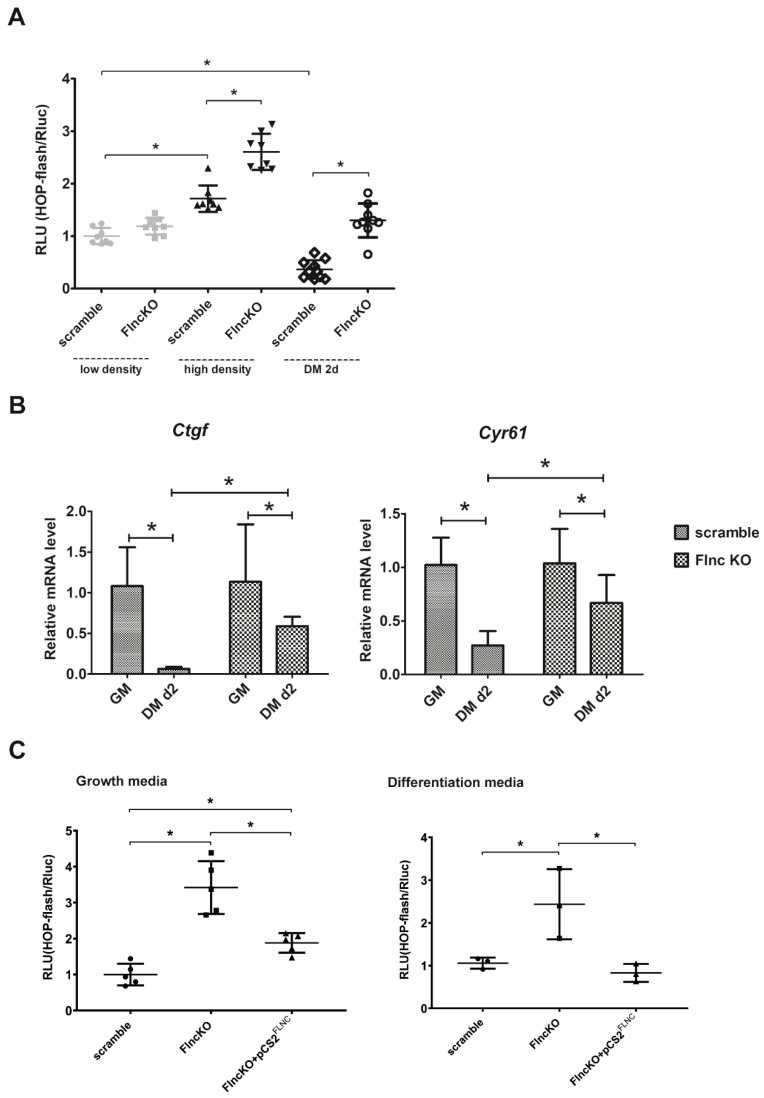
Loss of FLNC affects TEAD-YAP/TAZ activity in myoblasts during differentiation. (A) HOP-flash luciferase assay was performed in both under proliferating condition and during differentiation. In low density conditions no differences of TEAD-YAP/TAZ activity were obtained, whereas in high density *FlncKO* cells exhibit significantly greater increase of TEAD-YAP/TAZ activation. 48 h after changing of the growth media to differentiation media in both *FlncKO* and scramble cell line we observed significant decrease of TEAD-YAP/TAZ activity. (**B**) Expression of downstream YAP/TAZ-targeted genes were evaluated using qPCR. In scramble cell line mRNA expression of *Ctgf* and *Cyr61* become reduced 48 h after myogenic differentiation induction, when *FlncKO* cells exhibit weaker reduction of *Ctgf* and *Cyr61* expression under the same differentiation conditions. (**C**) Hippo pathway activity was restored in *FlncKO* cell line 24 h after add-back of FLNC using transfection with pCS2*^FLNC^* plasmid, when myoblasts were cultivated in growth media, and 48 h after induction of myogenic differentiation. For all * *p* < 0.05.

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
