# Peer review of "FLNC Expression Level Influences the Activity of TEAD-YAP/TAZ Signaling"

_genes, 2020, doi:10.3390/genes11111343_

Round 1
Reviewer 1 Report
The authors have provided a better characterisation of their knockout lines in this version mainly in terms of expression levels of Flna and Flnb in the Flnc knockout lines and expression of cytoskeletal markers. While there are some changes in the changes in the expression of Flna in the Flnc KO lines, I think the authors have clearly laid out what these differences are so as to make the interpretation of the results attributed to loss of Flnc clear. This point could be further clarified and strengthened in the discussion section.
Minor comments:
I am not sure I understand this sentence: "Surprisingly, we did not observe critical changes of F-actin network in myoblast cells in line with previously published data that loss of FLNA in variety of cell lines is also not associated with F-actin defects."
Author Response
Dear reviewer,
Thank you very much for the comment. By this sentence we wanted to mention, that due to loss of another filamin family member protein, FLNA, no changes of F-Actin organisation were found, the same as in our data, describing FLNC knock-out. Changes in text were made to clarify this sense (lines 290-291).
Reviewer 2 Report
The authors have satisfactorily addressed all my comments.
Author Response
Dear reviewer,
Thank you for the critical revision of our manuscript.
This manuscript is a resubmission of an earlier submission. The following is a list of the peer review reports and author responses from that submission.
Round 1
Reviewer 1 Report
In this paper, the authors examine the role of filamin in muscle cells. They generate a C2C12 murine myoblast cell line with reduced filamin expression and characterise the mutant line in terms of proliferation and differentiation. In addition they also examine the activation of hippo signalling pathway and its role in myogenic differentiation after the loss of filamin. Overall, the experimental aspect of the paper are well-controlled; however I have a few concerns about the interpretation of the data and the way it is presented.
Major concerns:
- Overall the authors have cited the appropriate literature. However, some of the results of the paper should be put in the context of known literature. For example, in figure 3 the authors showcase a very pronounced decrease in myogenic differentiation in FLNC-deficient C2C12 myoblasts. While reading the paper as it is written, it seems as though this is a novel finding, however, a quick look in the literature shows that this has been described before (Dalkilic I et al., Molecular Cell Biology, 2006), a paper that the authors have cited in the introduction. The authors should note that their observation is consistent with previous finding so as not to give an impression that this is first time this has been reported.
- I am bit unclear on the authors interpretation of the hippo signalling. The way I understand this, it seems that the loss of FLNC leads to high proliferation and loss in differentiation phenotype (a phenotype that has also been reported before). The authors in the discussion section then cite a number of papers which have stated that hippo signalling regulates cell proliferation and growth programs. They also mention that myogenic differentiation is inhibited at a high level of hippo signalling. Isn’t what is being observed simply a signature of the state the cell is in rather than a direct link between FLNC and the hippo pathway?
Minor concerns:
- For statistical analysis, please indicate the type of statistical test done.
Suggestions for future work.
While this is outside the scope of the current work, the authors could consider performing a global RNA-seq analysis on the filamin targeted cell lines. A global view would provide a much better understanding of the key cellular elements governing this process.
Reviewer 2 Report
Filamins are actin crosslinking proteins comprising of Filamin A, B and C. Filamin C is specifically expressed in skeletal and cardiac muscle tissue. Filamins are implicated in a large number of functions namely including cell–cell and cell–matrix adhesion, mechano-protection, actin remodeling, and various intracellular signaling pathways. Filamin C have been shown to interact with several key members including integrin, myotilin, FATZ-1 and myopodin and thereby implicated in maintainence of mechanical integrity of the muscle cells. In the present manuscript, the authors create a filamin C specific KO cell line and satisfactorily characterize them based on proliferation, cell migration and alteration of Hippo pathway. Although the data provided is interesting, the changes shown do not involve any possible mechanism for Filamin C mediated function in the cells. Moreover, the authors need to characterize the cell lines in a detailed way before ascertaining functional alternation to this cell line.
Following are my major suggestions:
- Generation of a CRISPR/Cas9 mediated KO lines are almost a routine and established procedure in research and hence the data can be shifted to a supplemental image.
Though a ponceau staining is provided, the authors still need to add in a loading control for the western blot shown.
Are filamin A and B expressed in these cells, if so, the authors need to show the levels of these proteins in their Filamin C KO cell line at various stages. Additionally, the authors need to blot for its known interacting partners and functional effectors to show their levels.
Since Filamin C is an actin-interacting proteins and aids in cell-cell contacts, the authors need to thoroughly characterize the KO line based on:
- Stress fiber compositions (Dorsal, ventral, transverse arcs)
- Lamellipodial compositions and dynamics
- Filapodial characteristics and dynamics
- Phenotype of overall actin filament meshwork.
- In Fig 2, it is most likely that the FLNC KO cells lack contact inhibition and are therefore outgrowing the control cells. To establish this the authors, need to at least show that the cellular size of the KO cells is similar to the control cells.
Moreover, owing to its reduced migration phenotype, the authors need to characterize the lamellipodial and filopodial dynamics in these KO cells along with evaluation of the known factors that aid cell migration (Formin proteins, Arp2/3 complex etc.).
Fig 1C could benefit from the better image as in the micrograph provided the initial wounding area of the control and the FLNC KO line is not similar. Moreover, the quantification provided do not effectively reflect the micrograph provided.
- For most of the experiments in Fig 2 and 5, the authors need to perform rescue experiments with FLNA, FLNB and FLNC to ascertain the function shown to FLNC. Also, it would be useful if the authors can provide some sort of explanation in the discussion on the results, they obtain in fig 5.
Minor comments:
- Fig 1 can be shunted to supplementary figures.
- Fig 3 and 4 can be combined into a single figure.
- Micrographs in Fig 2 and Fig 3 should have scale bars.
Round 2
Reviewer 1 Report
The authors have added a few sentences with regards to my previous comments, however the paper is still weak in terms of its experimental data backing the the link to Hippo signalling pathway. This can be solidified best with approaches like RNA-seq experiments and while I understand that that this may not be within the scope of the manuscript at the current stage, the authors should still discuss the limitations of their approach and discuss the further steps that they hope to take in the future to solidify their hypothesis.
Reviewer 2 Report
In the revised manuscript the authors have not addressed most of the concerns raised in the previous versions. While it is understandable that a few of the experiments might be out of scope for the present manuscript, some of the experiments asked for is required for the establishment of the current manuscript and functional role of FLNC. Following are my concerns:
- It is not required to hunt for loading controls of similar molecular weight as that of FLNC, actin, tubular, GAPDH etc can be used as a loading control for the fig. Ponceau staining cannot be agreed upon as a loading control.
- The levels of FLNA and FLNB in the FLNC KO cells must be provided to ascertain the phenotype to FLNC KO and not elevated expression of the other FLNC.
- Without a proper characterization of a KO line, its functional relevance cannot be ascertained. The authors need to provide certain level of characterization of the KO lines (stress fibers, lamellipodia, filopodia) to establish the functional consequence.
- I believe that the assay in Fig 1C was done more than once, in the view of which the authors need to provide a clear example in fig 1C.
- To ascertain a function based on a KO line, the authors either need to show it in KO lines established from different clones, or perform rescue experiments. In the present manuscript the authors should at least try to perform rescue experiments with FLNC to establish the results.